# On the Automatic Generation and Simplification of Children's Stories

**Maria Valentini**[▽]     **Jennifer Weber**[▽]     **Jesus Salcido**[▽]     **Téa Wright**[▽]
**Eliana Colunga**[▽]     **Katharina von der Wense**[▽◇♠]
[▽]University of Colorado Boulder
[◇]Johannes Gutenberg University Mainz
{first.last}@colorado.edu

## Abstract

With recent advances in large language models (LLMs), the concept of automatically generating children's educational materials has become increasingly realistic. Working toward the goal of age-appropriate simplicity in generated educational texts, we first examine the ability of several popular LLMs to generate stories with properly adjusted lexical and readability levels. We find that, in spite of the growing capabilities of LLMs, they do not yet possess the ability to limit their vocabulary to levels appropriate for younger age groups. As a second experiment, we explore the ability of state-of-the-art lexical simplification models to generalize to the domain of children's stories and, thus, create an efficient pipeline for their automatic generation. In order to test these models, we develop a dataset of child-directed lexical simplification instances, with examples taken from the LLM-generated stories in our first experiment. We find that, while the strongest-performing lexical simplification models do not perform as well on material designed for children due to their reliance on LLMs, a model that performs well on general data strongly improves its performance on children-directed data with proper finetuning, which we conduct using our newly created child-directed simplification dataset.

## 1 Introduction

Large language models (LLMs), such as GPT-3 or ChatGPT, are able to produce stories that are far more coherent and fluent than stories generated by state-of-the-art models from even a couple of years ago, such as GraphPlan (Chen et al., 2021a) and Plan-and-Write (Yao et al., 2018). However, most of the already limited work on automatic story generation focuses on stories for an adult audience. Children's stories are not frequently a topic of interest, despite how crucial early literacy is to future success (Walker et al., 1994).

[♠]Formerly: Katharina Kann

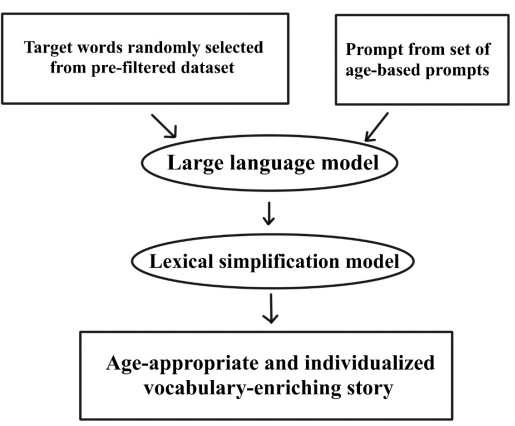

Figure 1: Simple example of a pipeline for the generation of educational children's stories.

As we will describe in more detail in Section 2, children's stories are important for both entertainment and education. Automatic story generation for children increases their potential for broader impact by making it possible to personalize stories, making them increasingly relevant for each individual child. As one example, tailoring stories to a specific child's interests could allow them to become more easily interested in reading, which could improve their literacy skills. Another possible application could be the teaching of specific target words to preschoolers via stories. In this paper, we will keep the latter use case in mind.

While, at first look, stories that have been generated by LLMs seem (and generally are) better than previous attempts, it has not been systematically evaluated if their children's stories adhere to what one would expect from the genre. In this paper, we focus on the *simplicity* of generated stories as measured by the age of acquisition of their individual words. Specifically, we assume that we are interested in generating stories to teach words with an age of acquisition (AoA) from 6 to 9 to children who are preschool-aged (around 2-5). Including

other complex words or concepts, however, can make story-based vocabulary learning more difficult, as they detract attention from target words and complicate the ascertainment of context clues. Thus, we want to include the target words in the stories, but we do not want any other words to have an AoA $\geq 6$.

For the first experiment conducted in this paper, found in Section 4, we use 3 different LLMs for generation and ask the following research questions: (RQ1) What is the simplicity of generated stories for different LLMs? (RQ2) How do different prompts, with different descriptions of the target age group, influence story simplicity for different models? We find that, in spite of LLMs' growing abilities for story generation, they struggle to control for age-appropriate simplicity; in comparison to our dataset of human-generated stories for the same demographic, the models' stories exhibit scores that are over 17% worse for all readability metrics tested (Flesch Reading Ease, Flesch-Kincaid Level, Gunning-Fog Index, and Automated Readability Index).

Motivated by those findings, in Section 5, we then turn towards simplification models. Simplification models have mostly[1] been developed for text directed at adults. Here, we investigate two leaders of the TSAR-2022 Shared Task on Multilingual Lexical Simplification (Saggion et al., 2023) in English, UniHD (Aumiller and Gertz, 2022) and UofM&MMU (Vásquez-Rodríguez et al., 2022), with regards to their ability to generalize to the challenging out-of-domain setting of children's stories. For this, we generate a new dataset of annotated instances for lexical simplification, using age of acquisition as a metric for identifying complex words and pulling examples from our newly created corpus of LLM-generated stories for human annotators to simplify. We then use this dataset to evaluate the models' performance on automatically generated children's stories. Our experiments show that, while UniHD performs considerably worse, achieving an accuracy of only 30.52% in comparison to the 42.89% accuracy it achieves on the TSAR-EN dataset for general lexical simplification, the ordinarily weaker UofM&MMU model is actually able to perform better on the child-directed dataset when finetuned, achieving an accuracy of 47.37% compared to its 28.95% on TSAR. We conclude that

simplification models trained on adult data are suitable to simplify automatically generated children's stories only when properly finetuned, i.e., the best-performing models for adult text do not generalize well without additional training, but fintuning is effective for our domain, even with limited data.

To summarize, our contributions are as follows: (1) We examine and compare the ability of several LLMs to adjust for age-appropriate levels of readability and lexical simplicity in automatically generated and individualized educational stories. (2) We explore how effectively state-of-the-art lexical simplification models perform in the domain of children's stories in order to supplement the inability of LLMs to sufficiently adapt their vocabularies. (3) We provide a public dataset for the simplification of child-directed text in order to promote the advancement of models for this purpose.

## 2 Background: Why Should LLMs Generate Children's Stories?

In child development, small early differences can compound into big long-term effects. One example of this is the relationship between early vocabulary size, literacy, and later academic achievement. Early vocabulary size is strongly related to reading ability in 2nd and 3rd grade (Walker et al., 1994; Fewell and Deutscher, 2004), and even when controlling for vocabulary size in Kindergarten, reading ability in 4th grade is associated with vocabulary growth through 10th grade (Duff et al., 2015). Although there is a recognition of this relationship, large gaps in vocabulary size persist into elementary school and beyond: e.g., Biemiller and Slonim (2001) estimate the vocabulary of 4th graders in the lowest quartile to be less than half the size that of 4th graders in the highest quartile, and a similarly sized gap is found in empirically-based estimates throughout adulthood (Brysbaert et al., 2016).

Many vocabulary enrichment programs based on shared reading with a caregiver have been developed to try to address this gap, with mixed success. Early vocabulary interventions are generally based on storybook reading with a parent or teacher. A meta-analysis focusing on vocabulary intervention studies on children in pre-K and K concluded that although such interventions may increase oral language skills, they are not powerful enough to close the gap, even when implemented at this early age (Marulis and Neuman, 2010). Experts agree that intensive, individual-level interventions would be

---

[1]Exceptions exist but are relatively old and use since-deprecated technology, such as (De Belder and Moens, 2010-01-01) and (Vu et al., 2014).

necessary to make a difference but acknowledge that the infrastructure investment required for something on that scale would be substantial (Suskind et al., 2013).

With recent improvements in LLMs, the concept of using natural language processing techniques to automate the child-by-child customization of educational materials has become increasingly realistic. In particular, state-of-the-art models have proven to be effective for generating individualized reading materials without the labor cost of having a human modify them by hand, making this process more practical in lower-resource settings facing educational discrepancies. Developing efficient pipelines for personalized vocabulary-enriching story generation is one way ito address this so called "achievement gap" in early childhood education, providing opportunities for individualized education where they otherwise may not be available.

There are several challenges that should be addressed in order to make the automatic generation of individualized educational children's stories a more feasible idea. One primary concern is ensuring that the stories are simple enough to be understood by the target demographic. Stories which are too complex or which contain too many unknown words could make understanding the meanings of their target words based on the context provided more difficult. Additionally, studies show that having fewer unfamiliar detractors allows children to focus on and retain target words more effectively (Horst, 2013). As such, it is important that automatic story generation models be able to control for the simplicity and readability of the stories outside of the words they intend to teach. Because of the potential personalized stories have for increasing both specific educational potential and overall engagement in reading, this paper focuses on how LLMs and lexical simplification models can be used for this purpose.

## 3 Related Work

### 3.1 Automatic Story Generation

The automatic generation of stories is a task which has seen vast changes in its methodology with the recent advent of LLMs, requiring increasingly less human intervention to mimic human-written stories. Even fairly recent automatic story generation models such as that by Li et al. (2013), GraphPlan (Chen et al., 2021b), and the Plan-and-Write system (Yao et al., 2019) focused much of their effort on setting up scaffolding for the story ahead of time in order to ensure the coherence of the eventual generation model. With LLMs such as ChatGPT (OpenAI, 2022) and Bard (Google, 2022), however, systems for automatic story generation can begin to rely less on this strategy of planning story structure ahead of generation.

More modern approaches that implement LLMs such as Future Sight (Zimmerman et al., 2022) and MEGATRON-CNTRL (Xu et al., 2020) allow for more creative modification during and after generation. Wordcraft (Yuan et al., 2022), for instance, allows end users to collaborate with OpenAI's GPT-3 in order to continually modify stories throughout the process of their generation.

One facet of story generation that has shown especially prevalent applications of late has been the generation of children's stories. Early childhood literacy has proven to be a significant indicator of a child's future academic success (Walker et al., 1994), so emphasizing generation for childhood literary advancement is one way in which modern NLP techniques can be especially impactful. Though many story generation systems have aspects that can likely be transferred to children's stories, few studies intentionally focus their generation systems directly at children.

### 3.2 Prompt-tuning

A topic that has seen increasing amounts of attention as LLMs have become more and more popular is prompt-tuning, or the study of how the modification of prompts to LLMs affects their output. The concept of prompt-tuning in the sense in which it is used in this paper was originally proposed in Lester et al. (2021), who propose the idea of modifying prompts for GPT-3 in a way that affects the model's results quantifiably, similar to how parameter tuning affects ordinary machine learning models. Other papers such as Hu et al. (2022) and Gu et al. (2022) reaffirm the effectiveness of prompt-tuning for LLMs, demonstrating that it can have an even more significant effect than traditional parameter finetuning. This paper does not place a large emphasis on prompt-tuning, but we do perform prompt-tuning on a smaller scale to investigate how specifying different target groups affects the simplicity of generated stories.

### 3.3 Lexical Simplification

Lexical simplification describes the process of identifying words which are too complex for some

target demographic and replacing them with synonyms which are easier to understand. Lexical simplification (LS) has been a commonly studied NLP tasks for several years, and early models such as De Belder and Moens (2010-01-01) and Vu et al. (2014) highlighted in particular the applications of lexical simplification for children or for non-native speakers (Petersen and Ostendorf, 2007). More recent models such as LSBert (Qiang et al., 2020), which was created by finetuning the BERT masked language model on the task of lexical simplification, focus instead on the task of general LS, bringing up the question of whether these more modern, higher-performing models can be generalized to work as effectively on children-directed text.

Even more recently, the TSAR-2022 Shared Task on Multilingual Lexical Simplification (Saggion et al., 2023) has drawn more attention to the improvement of LS models such as the winning UniHD model (Aumiller and Gertz, 2022). This shared task also led to the development of the ConLS system (Sheang et al., 2023), which is a modified version of LSBert created after the TSAR shared task and obtains state-of-the-art results on the TSAR-2022 dataset.

## 4 Experiment 1: Assessing Readability

First, we investigate the readability of automatically generated stories for 1) different models and 2) a variety of prompts. We generate a total of 250 stories for each model: 50 per model–prompt combination.

### 4.1 Models

**InstructGPT** InstructGPT is a group of GPT-3 models finetuned via reinforcement learning from human feedback. Trained on 1.3 billion parameters, it was released by OpenAI in January 2022 as part of its series of generative pre-trained transformer (GPT) models. These models use data gathered by crawling the internet to predict how a series of text tokens should be completed (Brown et al., 2020). The InstructGPT series of models are unique from other GPT-3 models in their intentional alignment with their purpose, which is completing text given a natural language instruction, as opposed to the inherent misalignment faced by models that just aim to predict statistically what word(s) should come next (Ouyang et al., 2022). Specifically, this experiment implements OpenAI's Text-DaVinci-003 model, which can be used at the cost of $0.0200

per 1000 tokens.

**ChatGPT** ChatGPT, or GPT-3.5-Turbo, is another model created by OpenAI using reinforcement learning from human feedback, released in November 2022 (OpenAI, 2022). Unlike Instruct-GPT, ChatGPT is finetuned in a supervised setting by using human-created AI assistant dialogue samples, making it more equipped for dialogue. It is currently free to use as a part of its research preview, but its code is not yet publicly available.

**Vicuna** Vicuna is a LLM created by finetuning the open-source LLaMa (Meta, 2023) on the ShareGPT dataset of user conversations. According to preliminary evaluations done using OpenAI's GPT-4, Vicuna is able to achieve 92% of the performance of ChatGPT (Chiang et al., 2023). Vicuna has the advantage over both GPT-3 and ChatGPT, however, that it can be used at no cost and has publicly available code. It is included in this study to test the capabilities of openly available LLMs to generate age-appropriate educational stories for preschoolers. This experiment uses the version of Vicuna with 7 billion parameters, which is built off of LLaMa's 7 billion parameter model.

### 4.2 Datasets

**Age of Acquisition Data** The dataset used to identify words in the model-generated stories that should be simplified in order to reduce their complexity and increase their educational potential is the English Lexicon Project's Age of Acquisition dataset (Kuperman et al., 2012). This dataset consists of over 31,000 words along with the estimated average age at which they are learned based on crowd-sourced data collected by researchers at six universities.

**Books for Preschoolers** In order to have a set of stories with which to compare the ones generated by the above-described LLMs, we use the Books for Preschoolers dataset (BfP) from Wiemerslage et al. (2022). It consists of 1026 human-written stories intended for children ages 2-5. Of all the words in this corpus, 88.61% can be found in the Age of Acquisition dataset (Kuperman et al., 2012) which is used to test the simplicity of the stories generated by the LLMs. Among the words that could be found in the AoA data, 83.08% were below the age threshold we compare to in computer-generated stories, which is 6. The stories included in BfP are commercially-available, professionally-written

picture books (i.e., books that have illustrations in every single page) intended to be read to preschoolers or by early readers, such as *Good Night*, *Moon* and *The Very Hungry Caterpillar*. Transcribed stories in the corpus contain an average of 52 sentences and 9.4 words per sentence. The authors themselves or the publishers designated the stories' age range that qualifies them for inclusion in this dataset.

## 4.3 Prompts

With our prompts, we aim to provide the model with the necessary information for generation of stories around the target words. In addition, we want to encourage simplicity, i.e., stories that are easily understandable by young children. We assume our target group consists of children aged 6 or younger.

We experiment with the following prompts:

- Write a story for a **preschooler** containing the following words: w1, w2, w3, w4, w5

- Write a story for a **3-year-old** containing the following words: w1, w2, w3, w4, w5

- Write a story for a **4-year-old** containing the following words: w1, w2, w3, w4, w5

- Write a story for a **5-year-old** containing the following words: w1, w2, w3, w4, w5

- Write a **children's** story containing the following words: w1, w2, w3, w4, w5

**Target Words** With the target demographic of preschool-aged children in mind, we select the target words for our LLM-generated stories from words in the AoA dataset with ages of acquisition between 6 and 9. Starting with all the words in this age range, we go through a specific filtering process that includes steps such as removing adverbs and words tagged with more than one part of speech, avoiding multiple words derived from the same lemma, and removing words with missing or low concreteness scores. The target words are then reviewed by multiple annotators and scored in three categories: learnability, imageability, and appropriateness. The guidelines relating to these categories (as they were presented to the annotators) are included in Appendix A. At this point, only the highest scoring words in each category are kept, resulting in a list of 150 nouns, 50 verbs, and 50 adjectives. The complete list of target words can also be found in Appendix A.

## 4.4 Metrics

Currently, automatic readability metrics are limited and largely consist of ones that are significantly outdated (Vajjala, 2022). These measures, although well-established and widely used, are coarse oversimplifications of language use. Since we are using several different measures consistently between the human- and computer-written stories, however, we believe these imperfect measures serve as a good starting point for comparison against the BfP corpus. As such, to judge the simplicity of the stories generated in Experiment 1, we use the following metrics.

**Average Age of Acquisition** We go through the stories generated by each model and find each word's age of acquisition. We then take the average from all of these words so we can judge the ability of each model to simplify their lexicon to reflect that of their target demographic.

**Average Highest Age of Acquisition** We check each story generated by a model and find the word in it that has the highest age of acquisition. We then take the average of these scores for each model to judge each model's ability to avoid using words in stories which are too complex for their target demographic.

**Readability Scores: Flesch Reading Ease** We use readability scores to judge the relative ease of reading each of the models' stories. The Flesch Reading Ease score is calculated using the following formula: Reading Ease = 206.835 – (1.015 x Average Sentence Length) – (84.6 x Average Syllables per Word).

**Readability Scores: Flesch-Kincaid Grade Level** Another readability metric we use to test the difficulty of reading each story is the Flesch-Kincaid Grade Level, which is computed via the following formula: Flesch-Kincaid Grade Level = (0.39 x Average Sentence Length) + (11.8 x Average Syllables per Word) - 15.59.

**Readability Scores: Gunning-Fog Index** Another readability metric we use is the Gunning-Fog Index, calculated as follows: Gunning-Fog Grade Level = 0.4 x (Average Sentence Length + Percentage of Hard Words), where "hard words" are defined as words with three or more syllables that are not (i) proper nouns, (ii) combinations of easy words or hyphenated words, or (iii) two-syllable verbs made into three with -es and -ed endings.

| Model | Prompt | Average AoA | Highest AoA | % Valid | % Appropriate |
|---|---|---|---|---|---|
| **InstructGPT** | **Preschool** | 4.75 | 9.15 | 90% | 0% |
| | **3-year-old** | 4.71 | 8.94 | 94% | 0% |
| | **4-year-old** | 4.72 | 9.21 | 94% | 0% |
| | **5-year-old** | 4.69 | 9.2 | 92% | 0% |
| | **child** | 4.81 | 9.67 | 96% | 0% |
| **Vicuna** | **Preschool** | 4.74 | 9.7 | 54% | 0% |
| | **3-year-old** | 4.64 | 9.57 | 58% | 0% |
| | **4-year-old** | 4.69 | 8.95 | 50% | 0% |
| | **5-year-old** | 4.66 | 9.34 | 52% | 0% |
| | **child** | 4.76 | 10.23 | 48% | 0% |
| **ChatGPT** | **Preschool** | 4.62 | 8.94 | 94% | 0% |
| | **3-year-old** | 4.64 | 9.01 | 96% | 0% |
| | **4-year-old** | 4.63 | **8.76** | 94% | 0% |
| | **5-year-old** | 4.67 | 9.24 | 98% | 0% |
| | **child** | 4.74 | 9.62 | 98% | 0% |
| **BfP** | | **4.6** | 8.87 | **100%** | **4.78%** |

Table 1: Full Experiment 1 results by model and prompt used. *Highest AoA* refers to the average highest age of acquisition. *% Valid* refers to the percent of stories including all desired target words, and *% Appropriate* is the percent of stories containing only words with AoA $\leq 6$. The human-written BfP corpus is included for comparison.

**Readability Scores: Automated Readability Index** The final readability metric we use is the Automated Readability Index, whose formula is:

$$4.71 \times \left(\frac{\text{characters}}{\text{words}}\right) + 0.5 \times \left(\frac{\text{words}}{\text{sentences}}\right) - 21.43$$

**% of Valid Stories** We further look at the % of valid stories to judge the models' ability to adhere to the prompts assigned. Stories are considered invalid if they are missing one or more of the assigned target words.

**% of Age-Appropriate Stories** Last, we compute the % of age-appropriate stories to judge the models' ability to adhere to the specified age group. Stories are considered invalid if they contain words with an age of acquisition higher than 6.

### 4.5 Results

Some key results for our first experiment are shown in Table 1. After running the above-described experiment, we find that although LLMs are generally able to simplify the *average* word difficulty in their stories to age-appropriate levels, they are unable to avoid including some words with age of acquisition levels significantly higher than their target demographic. Further, we find that none of the 750 total generated stories stayed within the age range of 6 or younger. Though it is common for some children's stories to contain words that are more complex, others refrain from using such words in order to cater to their target demographic (e.g., 49 stories in the Books for Preschoolers dataset). Having this

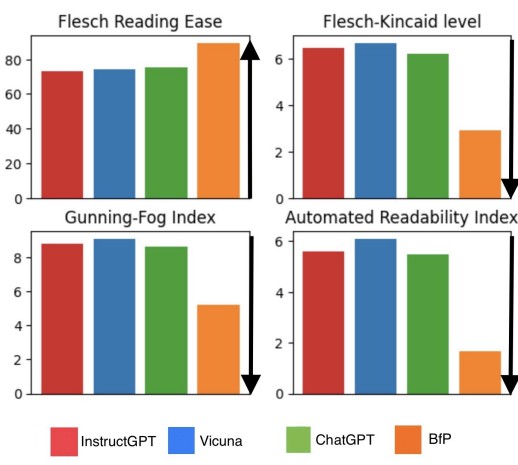

Figure 2: Readability metrics for each model, as well as the Books for Preschoolers human-written dataset (BfP). Note that the direction of the arrow on the side of each chart indicates the direction of improved readability.

ability is a key difference between computer- and human-generated stories, highlighting an area in which our automatically generated stories could use improvement.

In terms of readability, the stories generated by our models score significantly worse than those in the BfP dataset. While the average Flesch Reading Ease (FRE) in the BfP dataset is approximately 89.37 (out of 100), the average FRE among the stories generated by the models is only 74.22. Contrarily, for the Flesch-Kincaid Level (FKL) and

Gunning-Fog Index (GFI), a lower score indicates a body of text being easier to read. In the FKL and GFI metrics, the BfP stories score on average 2.9 and 5.23, respectively. Among the stories generated by the LLMs, meanwhile, these scores average out to 6.44 and 8.87. Full results of this readability analysis can be seen in Figure 2. With regard to our second research question, we find that age-specific prompt-tuning has little effect on the simplicity of children's stories generated by LLMs.

## 5 Experiment 2: Story Simplification

The results shown in Section 4.5 demonstrate a serious need for improvement concerning age-appropriate simplicity in automatically generated children's stories. Thus, in our second experiment, we examine to what extent current state-of-the-art lexical simplification models generalize to the domain of children's stories, as well as how this could be beneficial in the automatic generation of educational children's stories by LLMs.

### 5.1 Dataset Creation

To the best of our knowledge, there are currently no datasets for lexical simplification focused on children. As such, the first step in our second experiment is to create such a dataset. We are able to do this by using the corpus of children's stories created in Section 4, identifying complex words using the Age of Acquisition dataset, and having human annotators identify simpler synonyms for these complex words (in cases where such synonyms exist). In the annotation process, each instance consists of the sentence from which the complex words was taken, as well as the complex word itself. Annotators are tasked with finding a simpler synonym for each instance that could replace the complex word in the sentence without changing the sentence's meaning. Each instance is then reviewed by two additional annotators, a professor and PhD student in NLP, to ensure that the meaning of the sentence is retained. Only instances deemed to be valid are kept, meaning all annotators agree the sentence's meaning was unchanged and the newly suggested synonym has a lower age of acquisition than the original word. In total, we annotate 750 instances randomly selected from our corpus of LLM-generated stories. After filtering out instances deemed to be invalid, our final dataset consists of 315 simplification examples. We refer to this dataset as our **C**hild-**D**irected **S**implification

dataset, or CDS for short.[2]

### 5.2 Models

We use two of the three best systems of the TSAR-2022 Shared Task on Multilingual Lexical Simplification to test the ability of lexical simplification models to generalize to the domain of children's stories: UniHD (Aumiller and Gertz, 2022) and UofM&MMU (Vásquez-Rodríguez et al., 2022). In addition to examining the ability of state-of-the-art lexical simplification models to simplify our computer-generated children's stories, we also experiment with LLMs for the task to see if they outperform any models designed specifically for simplification.

**UniHD**   UniHD's model is created using an ensemble of six different configurations/prompt combinations from GPT-3. Its results are generated by calculating an aggregate ranking of the outputs of its different GPT-3 configurations and prompts. It demonstrates state-of-the-art performance in the area of lexical simplification, with an accuracy score over 25% higher than LSBert (Qiang et al., 2020), which is regarded as one of the most popular and effective LS models to date.

**UofM&MMU**   While UofM&MMU's performs considerably lower on accuracy on the TSAR dataset than UniHD does, it can be finetuned with additional data. Based on the BERT masked language model, the UofM&MMU model goes through three distinct steps in its simplification process. The first involves candidate generation based on different prompt templates to be provided to BERT. The second finetunes BERT and subsequently ranks and selects candidates. Finally, the candidates are post-processed in order to filter out noise and remove any antonyms that may appear. For this model, we are able to finetune using a training set consisting of 70% of the CDS dataset. The remaining 30% is used as a test set.

**Vicuna, ChatGPT, InstructGPT**   We further experiment with the three LLMs used in our Experiment 1. We use the following set of prompts to get data for our full range of metrics:

- Name a simpler synonym that could replace the word [*word*] in the following sentence: [*sentence*]

[2]The complete dataset can be found at https://github.com/mariavale/CDS.

- Name two simpler synonyms that could replace the word [*word*] in the following sentence: [*sentence*]

- Name three simpler synonyms that could replace the word [*word*] in the following sentence: [*sentence*]

### 5.3 Metrics

For Experiment 2, we use three metrics to measure the performance of the simplification models.

**Accuracy**   We define accuracy as the ratio of instances in which the top-ranked candidate generated by the model is equal to the top synonym chosen by human annotators.

**Simplification Validity**   This metric represents the ratio of instances where the model chooses a top candidate with a lower age of acquisition than the original word.

**Accuracy@k**   This is the ratio of instances in which at least one of the top-$k$ candidates generated by the model is equal to the top synonym chosen by human annotators. We calculate this with $k = 2$ and $k = 3$.

### 5.4 Results

Full results of our second experiment with regard to the tested simplification models can be found in Table 2. Upon running the UniHD model on our child-directed simplification dataset, we find that the performance of the model is significantly worse than it is on the TSAR shared task English dataset. In terms of accuracy, the model is able to generate a top candidate equal to the one selected by human annotators 30.52% in comparison to 42.89% in the adult-directed dataset. Regarding a pure LM employed with prompting, we find that, with accuracy scores lower than all but one of the other models and validity scores significantly lower than any of them, LLMs are not effective tools for this task.

However, we find different results for the UofM&MMU model in combination with finetuning. On the original TSAR-EN dataset, UniHD outperforms UofM&MMU by over 15% accuracy. Without finetuning, the UofM&MMU model also performs significantly worse on the child-directed dataset, with an accuracy score of just 8.42%. After being finetuned with a portion of the CDS dataset, however, the model is able to score even better than the best-performing model does on the TSAR

dataset, predicting the top human-selected substitution with 47.37% accuracy. This demonstrates that finetuning can result in ordinary lexical simplification models being able to generalize to simplify child-directed text and that even better results could be achieved if better-performing models allow for the same level of finetuning.

We conclude that, while LLMs and LLM-based models struggle to simplify children's stories, models which allow for finetuning on domain-specific data can perform as well as or even better on children's stories than they do on adult-directed corpora. In terms of our overall pipeline, we conclude that it is in fact plausible to generate and simplify children's stories using LLMs for generation and finetuned lexical simplification models to simplify overly complex words.

## 6 Conclusion

In this paper, we investigate the ability of several current LLMs to generate age-appropriately simplified stories for children, as well as an examination of how modern lexical simplification models generalize to the domain of children's stories to enhance their educational potential. We demonstrate that, in spite of their growing capabilities, modern LLMs are unable to generate children's stories with age-appropriate simplicity, particularly in comparison to their human-written counterparts. Because of these shortcomings found in the automatically generated stories, our second experiment (Section 5) focuses on whether or not ordinary lexical simplification models generalize to the domain of children's stories, due to the lack of current LS models that focus on children-directed corpora. We find that some models which are ordinarily lower-performing than their LLM-powered counterparts have the potential to perform well in the domain of simplifying child-directed text, when properly finetuned.

Over the course of our experiments, we further create a corpus of vocabulary-driven LLM-generated children's stories as well as an annotated lexical simplification dataset, CDS, intended specifically for the domain of children's text and using examples taken from this above-mentioned automatically generated stories. We provide these datasets publicly in order to promote further research in this area.

In future work, we hope to further improve the automatic generation of customized children's sto-

| TSAR-EN | | | | |
|---|---|---|---|---|
| | Accuracy | Validity | Accuracy@2 | Accuracy@3 |
| **UniHD** | **0.429** | 0.381 | **0.611** | **0.686** |
| **UofM&MMU** | 0.290 | 0.370 | 0.453 | 0.531 |
| **UofM&MMU(Finetuned)** | 0.290 | **0.467** | 0.453 | 0.531 |

| CDS | | | | |
|---|---|---|---|---|
| | Accuracy | Validity | Accuracy@2 | Accuracy@3 |
| **UniHD** | 0.305 | 0.616 | 0.368 | 0.379 |
| **UofM&MMU** | 0.084 | 0.716 | 0.147 | 0.189 |
| **UofM&MMU(Finetuned)** | **0.474** | **0.937** | **0.537** | **0.600** |
| **InstructGPT** | 0.111 | 0.394 | 0.238 | 0.248 |
| **ChatGPT** | 0.263 | 0.610 | 0.378 | 0.422 |
| **Vicuna** | 0.140 | 0.438 | 0.219 | 0.254 |

Table 2: Full results of Experiment 2. Comparison of the UniHD and UofM&MMU models' performance on the adult-directed dataset (TSAR-EN) and all tested models' performance on our dataset (CDS). *Validity* refers to the model predictions' simplification validity for each dataset.

ries by adding models for other tasks to our generation pipeline, such as one that can detect coherence errors or one that can improve readability.

## Limitations

We use a limited number of LMs and simplification models in our experiments, and the number of prompts we explore is also rather small. Thus, while our experiments feature state-of-the-art models, we cannot exclude with absolute certainty that other models or prompts might lead to different results. Our dataset is small as well, and it could be improved with the help of more annotators. Future research could include the use of more workers to create a significantly larger dataset, potentially through the use of gamification for data collection.

## Ethics Statement

Regarding the ethical considerations for this study, we find that the harms are minimal due to the relatively confined nature of the experiments; all annotations were performed voluntarily and with consent. Potential benefits of this study include the advancement of research in the area of child-directed lexical simplification and improvements in efficiency for the creation of personalized educational material for young children. Though the results of this research are eventually intended for the demographic of children, no vulnerable populations were involved in this study up to this point. If automatically generated stories are given or read to children, it is important to verify in advance that they are safe for the target population, as current models cannot guarantee this.

## Acknowledgments

We thank the anonymous reviewers for their helpful comments. This research was supported by the NSF under grant IIS 2223917. The opinions expressed are those of the authors and do not represent views of the NSF.

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

# A   Appendix

## A.1   Annotator Guidelines

**APPROPRIATENESS**   Rate each word with respect to how appropriate they are for children. A HIGH appropriateness (5) word will be totally fine for a preschooler; a LOW appropriateness (1) word should NOT appear in a story for preschoolers.

**LEARNABILITY**   Rate each word based on how likely you think a preschooler is to be able to learn it from a story. HIGH learnability (5) words should be easy to learn from a story; LOW learnability(1) words would be nearly impossible to learn from a story.

**IMAGEABILITY**   Rate each word with respect to their imageability. High imageability (5) words will easily evoke a mental image in your mind; LOW imageability (1) words will evoke a mental image with difficulty or not at all.

## A.2   Full List of Target Words

accordion, acrobat, almond, anteater, antelope, anthill, bakery, bandanna, banjo, beagle, billboard, blizzard, bobcat, bookmark, bookshelf, bouquet, bugle, campground, cashew, catcher, cavern, chandelier, cheetah, chef, chimpanzee, clipboard, cobweb, collie, comet, confetti, cookbook, cottage, cowbell, crater, crescent, cricket, cyclist, denim, desert, dome, doorknocker, dumbbell, earmuff, earplug, earring, easel, eggplant, elk, ferryboat, fireplace, flute, forearm, fountain, gator, glacier, gnome, golf, grove, gutter, hairnet, hammock, headstand, hedgehog, hexagon, hiker, hourglass, iceberg, iguana, island, jaguar, jersey, kayak, kiwi, lantern, lifeboat, limousine, llama, lobster, locker, macaw, mansion, maze, microwave, mole, moss, mountaintop, museum, musician, newt, nostril, orchard, pelican, petunia, pinwheel, piranha, platypus, pompom, poncho, propeller, receipt, rink, rocker, sardine, sax, sequin, shrimp, shrub, sibling, skillet, skylight, skyscraper, sloth, snowshoe, songbird, sparrow, spatula, speck, spotlight, squid, stadium, stairwell, statue, stethoscope, stopwatch, suitcase, tangerine, taxicab, tennis, thimble, thunderstorm, tightrope, tongs, tortilla, toucan, trolley, trombone, trouser, tuba, tulip, tumbleweed, tusk, tutu, vase, violet, violin, visor, volleyball, warthog, wreath, xylophone, bald, bearded, beige, blond, blurry, breakable, bubbly, bushy, chalky, chilly, cloudless, crumbly, electric, feathery, floral, foamy, foggy, frosty, gooey, grassy, greenish, hatless, hilly, lilac, longhaired, lumpy, magenta, moonless, moonlit, mossy, plaid, prickly, puffy, reddish, seaside, sleeveless, slimy, smoky, starry, stormy, stretchy, sunless, sunlit, swampy, thorny, turquoise, undersea, wintry, wooded, wrinkly, applaud, awaken, bulldoze, curl, dangle, darken, decorate, deflate, deliver, dine, dotted, drove, enlarge, erupt, exhale, expand, fetch, flatten, halve, hover, illustrate, inflate, invite, jog, juggle, knit, magnify, masked, mimic, mow, munch, perform, recline, repaint, rotate, serve, sew, skydive, sniff, soak, squint, stumble, sunbathe, topple, unfold, unhook, unlock, unpack, unroll, unzip