# OpenReview forum: "On the Automatic Generation and Simplification of Children's Stories"
_EMNLP/2023/Conference — EMNLP 2023 Main_

### Official Review · Reviewer_6sAM · 2023-07-30

**Soundness:** 3

**Excitement:**

3: Ambivalent: It has merits (e.g., it reports state-of-the-art results, the idea is nice), but there are key weaknesses (e.g., it describes incremental work), and it can significantly benefit from another round of revision. However, I won't object to accepting it if my co-reviewers champion it.

**Missing References:**

Even though it may not be peer-reviewed yet, a very relevant paper to consider citing is this one:

Eldan and Li (2023). TinyStories: How Small Can Language Models Be and Still Speak Coherent English? https://arxiv.org/abs/2305.07759

**Paper Topic And Main Contributions:**

This paper presents a system for generating children’s stories, where the stories must meet certain vocabulary constraints based on the age-of-acquisition of words. The work examines whether selected LLMs produce stories that meet these constraints when prompted with instructions to do so. It finds that these particular LLMs tend to produce words that are above the desired age of acquisition for children’s stories, thus failing to meet the task requirements. To address this, the authors examine models that have been shown to be useful for the specific task of lexical simplification. They apply these models to the LLM-generated outputs to yield some lexical changes that satisfy the age-of-acquisition constraints.

**Questions For The Authors:**

Why did you select target words in the range of 6-9 for the age of acquisition when some of prompts mention younger ages (i.e. “write a story for a [3,4,5]-year-old”)?

Also, if you using this 6-9 range, why is a story considered invalid if it contains a word with an age of acquisition higher than 6?

Regarding the results in Table 1, how do the human-authored BfP stories compare with these numbers?

**Reasons To Accept:**

The overall applied objective of the work is well-motivated. The authors effectively argue for the practical benefit of the task: being able to quickly generate children’s stories, especially ones that can be customized to a particular child’s interests, could improve students’ reading ability.

This paper also benefits from defining the generation task through specific objective criteria: in particular, the authors use an existing resource (the English Lexicon Project’s Age of Acquisition dataset) to determine the lexical constraints for the stories. This enables the generated stories to be readily evaluated in terms of how well they respect these constraints.

**Reasons To Reject:**

The paper claims that its two-stage pipelined approach, where an LLM generates the initial story and a separate lexical simplification (LS) model is applied to this story, is effective for the task. While the paper demos this pipeline, the experiments do not show that this particular approach is favorable to any alternative approach. In particular, the paper does not compare its pipeline to an LLM-only approach or an LS-only approach, nor does it compare the LLMs and LS models to each other. These two components occupy different sections in the paper, and in each section different datasets and evaluation metrics are used. The paper assumes that the LLMs and LS models will receive different inputs: for LLMs, it’s an instruction prompt, while for the LS models, it’s an existing story. However, an LLM could similarly take an existing story as input (along with a prompt instructing the model to simplify it), which would enable it to be directly compared to the LS models. This should be included in the experiment to strengthen the paper's claims that LLMs are not sufficient for the task.

The paper presents a new dataset for the task, CDS, but its design and purpose is not adequately detailed. The paper utilizes an existing dataset of children’s stories, BfP. The LLM experiments utilize BfP for evaluation, while the LS experiments use CDS. More clarity is needed about the added value of CDS in relation to BfP. Is it that CDS explicitly annotates the lexical mapping between a story and its simplified version? What stories are used as the original source for CDS? Is it the human-authored BfP stories (as implied by Lines 509-511), or is it the LLM-generated stories (as implied by Lines 528-529)? If the task is to generate a simplified version of a complex-language story, then neither the BfP nor LLM stories seem well-suited to represent the “complex” side of the stories in a dataset for this task. This doubt could be resolved with some statistical description of CDS in terms of the frequency of words identified as complex in the original stories and thus replaced in the simplified versions. Line 531 indicates there are “315 simplification examples”, but it’s not clear whether an example is a single word or an entire story, and how many words per story are changed.

**Reproducibility:**

3: Could reproduce the results with some difficulty. The settings of parameters are underspecified or subjectively determined; the training/evaluation data are not widely available.

**Reviewer Confidence:**

4: Quite sure. I tried to check the important points carefully. It's unlikely, though conceivable, that I missed something that should affect my ratings.

**Typos Grammar Style And Presentation Improvements:**

Figure 2 refers to the human-authored stories as a “baseline”, which is not the best term since “baseline” typically refers to a weak model. Renaming this to simply “human” would be clearer.

Even though it’s explained in the text, it may be helpful in Figure 2 to add an up or down arrow to each of the three x-axis captions to indicate whether lower or higher scores are more desirable, since this differs between the three metrics.

---

> ### Author Rebuttal · Authors · 2023-08-29
>
> We really appreciate this review, and you bring up several good points that we certainly agree would improve the overall quality of the paper.
>
> Your mention of our second experiment lacking results regarding the use of an LLM for simplification is an excellent point. We did a fair amount of experimentation with LLMs before deciding we needed to explore other models, so it would definitely make sense to formalize this and include it in the paper to further demonstrate the soundness of our conclusions. As such, we present below the experiment we have now run, using several different prompt templates to test the models’ performance in the same ways as the other LS models. The prompts used were as follows:
>
> 1. *Name a simpler synonym that could replace the word [word] in the following sentence: [sentence]*
>
> 2. *Name two simpler synonyms that could replace the word [word] in the following sentence: [sentence]*
>
> 3. *Name three simpler synonyms that could replace the word [word] in the following sentence: [sentence]*
>
> These prompts were selected in order to get all metrics for comparison (accuracy@1-3 and validity). The table below shows the results of this experiment.
>
> | **Model**       | **Accuracy@1** | **Validity** | **Accuracy@2** | **Accuracy@3** |
> |-----------------|--------------|--------------|----------------|----------------|
> | **InstructGPT** | 0.111        | 0.394        | 0.238         | 0.248        |
> | **ChatGPT**     | 0.263        | 0.610        | 0.378         | 0.422        |
> | **Vicuna**         | 0.140        | 0.438        | 0.219         | 0.254        |
>
> With accuracy scores lower than all but one of the other models we tested and validity scores significantly lower than any of them, the results from even the best of these language models demonstrate that LLMs are not effective tools for this task.
>
> We also tried another line of full story prompting but found it was also not effective for our purposes. Even with our most detailed prompt:
>
> *Simplify the following story to improve its readability for preschool-aged children:*
> *[story]*
> *Try to use simpler words and not to shorten the story.*
> *However, do not remove the following words: [targets]*
>
> the LLMs were only able to generate valid simplifications that left all target words in the story less than 60% of the time, despite only decreasing the stories’ average age of acquisition by just over 0.02 years.
>
> Additionally, your suggestion of including more details to demonstrate the importance of the CDS dataset is a good one, and we would add these details into the next iteration of the paper. In regard to the specific questions you asked, we would first like to clarify that the dataset is composed of 315 individual simplification examples, each consisting of the word in need of simplification, the sentence the word comes from for context, and the simpler synonym that was chosen by annotators to replace it. These examples were selected randomly from sentences in the computer-generated stories that were flagged as containing one or more words above our target age of acquisition. We intentionally use the already fairly simple computer-generated stories in order to test how well models can simplify for young children, i.e. simplifying words that are already simple enough to not be replaced in adult-directed lexical simplification. Our hope is that having this domain-specific simplification dataset available will encourage future research in this area and improve upon current methods.
>
> In your questions section, you ask why we selected words with age of acquisitions ranging from 6 to 9. Since the goal is for these stories to support vocabulary growth, we wanted to select target words that most children in our target age range would not know. The AoA metric is just an estimate, and individually, children can learn vocabulary words with an AoA above their actual age as long as they encounter the words in their environment and they are presented accompanied by the scaffolding context of language that they do understand, so we want the rest of the story to be written in the style and at a level for our target age range. We also reference (Horst, 2013) in our background section, which mentions that a story containing other unknown elements aside from the novel target words negatively impacts children’s learning of these words. Therefore, if any other words in our stories (all language surrounding our 5 target words with higher ages of acquisition) had a higher age of acquisition as well, we considered the story to be too complex for the child and thus invalid.
>
> We also appreciate your suggestion for an additional paper to examine (Eldan & Li, 2023) and agree it has some valuable information closely related to our paper. The goal for the GPT-generated stories is quite similar to ours in that the model is instructed to refrain from using words that are too complex for a young child to understand. It seems that even with the more precise instruction demonstrated in this paper, however, the model still fails to contain its vocabulary to the desired age range. Using the smaller validation set to estimate the statistics of the overall TinyStories dataset, we find that the average age of acquisition for the stories is 4.46, and the average highest age of acquisition is 8.29. These do show slight improvement over ours, which have an average AoA of 4.66 and an average highest AoA of 9.54, but still highlights the need for further simplification in our use case.
>
> Finally, thank you for pointing out the lack of comparison with the human-written stories in Table 1. We did not include BfP in this table due to the fact that it only had one statistic as opposed to the five prompt-based statistics that are available for the model-written stories. We would like to modify our Figure 2, which compares the model-written stories with the human-written ones in BfP, to include this missing information on age of acquisition.

---

### Official Review · Reviewer_Psov · 2023-08-02

**Typos Grammar Style And Presentation Improvements:** Line 490
**Soundness:** 4

**Excitement:**

4: Strong: This paper deepens the understanding of some phenomenon or lowers the barriers to an existing research direction.

**Justification For Ethical Concerns:**

I don't see any ethical concerns for this work.

**Missing References:**

1.

AoA data (section 4.2) has the wrong reference. It cites Balota et al. 2007, (line 351 and again line 363). However that dataset does not have AoA,
it has lexical decision and naming latencies.
The dataset with 31K AoA ratings is probably this one:

Victor Kuperman & Hans Stadthagen-Gonzalez & Marc Brysbaert

Age-of-acquisition ratings for 30,000 English words

DOI 10.3758/s13428-012-0210-4





**Paper Topic And Main Contributions:**

This paper examines the capability of LLMs to generate stories at the appropriate age level (stories for age of about 5).
The reported work includes two studies.
One study focused on different ways of prompting LLMs  and measuring the vocabulary complexity of resulting outputs relative to a database of Age of Acquisition  (AoA) (vocabulary in English).
The results were also evaluated for age-appropriateness with classic readability formulae.
The second study looked at the ability of some recent lexical-simplification systems to simplify stories to the level required.



**Questions For The Authors:**

1.

The 'Books for preschoolers' (BFP) corpus is cited for  Wiemerslage et al., 2022.
However, that paper barely mentions it on two lines and gives no details at all (!!!).
The present manuscript provides very scant detail about this corpus, all it says is:

"It consists of 1026 human-written stories intended for children ages 2-5. Of all the words in this corpus, 88.61% can be found in the Age of Acquisition dataset"

Since this collection is central to the reported work, more details should be provided about the corpus.
Who wrote the stories? (professional writers?)  How long are the texts on average (word counts)?  Who was qualified to designate them to ages range 2-5?
Are those commercial publications or something else (e.g. free materials)?

Why almost 12% of words were not found in the AoA data?  - what kinds of words did not have AoA: exclamations? proper names? etc.
Can you provide percentages by such groups (e.g. proper names)?

Note: since the BFP corpus is not released and is not public, this can affect the reproducibility of the study,
However, I did not lower the reproducibility score for that (as releasing such data can be difficult).

2.

The use of the Flesch Readability Ease (FRE) is somewhat problematic. For FRE, the scale starts with 100 (and goes down), but 100 corresponds to US grade level 5  (see  https://en.wikipedia.org/wiki/Flesch%E2%80%93Kincaid_readability_tests),
which corresponds to ages 10-11, well above your targeted level of ages 2-5 or even 6-9. So, this measure does not have enough resolution or sensitivity to measure readability for the age levels of interest.

Moreover, the paper says (lines 475-478) that the average FRE in the BFP corpus is 89.37 (well below 100), which is at about beginning of 6th grade level, i.e. age 10-11.
Does that imply that the BFP corpus is actually not suitable for children ages 2-5?!!!  Or maybe the BFP corpus is suitable, and FRE is just an inadequate measure for this age?  Something is incongruous here.

3.

Section 5.1. Human annotators selected synonyms.  Only in the Limitations section we get to know that those annotators were crowd-sourced. More details need to be provided about the adequacy of those annotators to the task.

4.

There is some potential to be misleading.
In the Conclusions section, lines 639-645, the text says "...LLM...have the potential to perform very well in...".
When one reads that formulation (without the numbers), and most readers read only abstracts and conclusions, one might consider that LLMs perform well on this task.
But actually the better result is just 36% (line 603), which is nice for research, but is way far from 'well performing' for any applied use.






**Reasons To Accept:**

Overall the study is very interesting and even exciting. Really nice!

As the popularity of novel LLMs and their wide adoption are growing, the aspect of text complexity can be very important for educational uses.
As  study 1 shows, LLMs do not perform adequately, at least for  very young children.
As study 2 shows, simplification for very young children is improving, but is not quite solved yet.

The idea of using AoA as a measure for vocabulary is a good one.

The study with simplification is actually a different direction from study 1, but it is conceptually well-related to study 1,
so overall the paper is very good.

**Reasons To Reject:**

There are some inadequacies with some of the measures. But no reason to reject.

**Reproducibility:**

4: Could mostly reproduce the results, but there may be some variation because of sample variance or minor variations in their interpretation of the protocol or method.

**Reviewer Confidence:**

5: Positive that my evaluation is correct. I read the paper very carefully and I am very familiar with related work.

---

> ### Author Rebuttal · Authors · 2023-08-29
>
> Thank you so much for this thorough review of our paper! We appreciate the consideration put into your questions for us and think that adding these details into the paper would be a great way to improve it for publication.
>
> In your first question regarding the BfP dataset, we agree that it is not well described in our current submission of the paper. We would like to clarify that the stories included in BfP are commercially-available, professionally-written picture books (i.e., books that have illustrations in every single page) intended to be read to preschoolers or by early readers, such as *Good Night, Moon* and *The Very Hungry Caterpillar*. Transcribed stories in the corpus contain an average of 52 sentences and 9.4 words per sentence. The authors themselves or the publishers designated the stories’ age range that qualified them for inclusion in this dataset.
>
> Regarding the words in the BfP corpus that were missing from the Age of Acquisition data, only 30.02% of those missing could be found in the English dictionary, meaning 69.98% were either names, made up words, or other non-word entities. Of those that were found in the dictionary, the large majority were either proper nouns (e.g., February, Arizona, American), plurals of nouns whose singular forms are in the AoA data, or past tenses of verbs whose present forms are in the AoA data.
>
> Your statement addressing the age mismatch with the readability metrics we are using (specifically FRE) is also an excellent point. Given that the books contained in the corpus are all commercially available, professionally written stories published for this age group, we believe that using the imperfect readability measures across the board and in comparison to the BfP corpus as a control is a reasonable place to start. Thanks to your comment, however, and upon further examination of readability formulae, we were able to find a measure that has sensitivity for younger ages to test as well; the Automated Readability Index is designed for ages 5 and up, meaning it can more accurately test whether stories are fit for our 2-5 age range. We found similar results using this metric, with the BfP corpus scoring an average of 1.67 (around Kindergarten level) and our model-generated stories scoring an average of 5.72 (around 4th grade level). Full results are shown below.
>
> | **BfP (Baseline)** | **InstructGPT** | **ChatGPT** | **Vicuna** |
> |--------------|--------------|----------------|----------------|
> | 1.67         | 5.60         | 5.46           | 6.10           |
>
> We also agree that our current manuscript is lacking in details relating to the annotations used for our study. As we mentioned to Reviewer 1, we would like to clarify that for each example to be added to the newly created CDS dataset, the top answer had to be agreed upon by 3 or more annotators, two of which always included a professor and PhD student working in the field of Natural Language Processing.
>
> Finally, thank you for your comment on the wording of our conclusion; we agree that this could potentially be misleading and would definitely prefer to present as accurate a picture of our results as possible.

---

### Official Review · Reviewer_Si5t · 2023-08-06

**Soundness:** 4

**Excitement:**

3: Ambivalent: It has merits (e.g., it reports state-of-the-art results, the idea is nice), but there are key weaknesses (e.g., it describes incremental work), and it can significantly benefit from another round of revision. However, I won't object to accepting it if my co-reviewers champion it.

**Missing References:**

- For resources and ATS work relative to children, there are some papers focusing on dyslexic children, such as Rello et al., 2014 or Gala et al., 2020.
Gala, N., Tack, A., Javourey-Drevet, L., François, T., & Ziegler, J. C. (2020). Alector: A parallel corpus of simplified French texts with alignments of misreadings by poor and dyslexic readers. In Proceedings of the 12th Language Resources and Evaluation Conference (pp. 1353-1361).
Rello, L., Baeza-Yates, R., & Llisterri, J. (2014). Dyslist: An annotated resource of dyslexic errors. In Conference on Language Resources and Evaluation LREC-2014, page 1289–1296, Reykjavick, Island.

**Paper Topic And Main Contributions:**

This article tackles two related research questions. On the one hand, it aims to determine whether LLMs are capable of generating stories for young children (2 to 6 years old) in which the age of acquisition of words and the readability of these stories is appropriate. On the other hand, they investigate the use of Lexical Simplification (LS) to compensate for the weaknesses of LLMs at the lexical level. The authors found that three LLMs (including ChatGPT) are not able to generate children stories at an appropriate level of difficulty, which is an interesting result to dampen the current trend to believe that “every task should be done prompting an LLM”. In addition, they apply state-of-the-art LS model to the generated texts with the aim of simplifying lexical load of these stories. They show that a model that can be fine-tuned performs better than an LLM-based model for children stories. Finally, a third contribution consists in a dataset of 315 sentences in which a complex word (as regards AoA) has been simplified by experts.

In my opinion, these three contributions are valuable and fit for EMNLP. Automatic generation of stories of different difficulty levels is also an area that is bound to flourish over the next few years, and this article is a good milestone. In addition, the paper reads well, is based on a use case that is well-motivated with theoretical work from various fields, and reports experiments based on a solid methodology. There are, however, a few shortcomings that prevent me from being completely enthusiastic about this paper.


**Questions For The Authors:**

- In the use case presented at pages 2-3, it is not clear for me who will read the text to children aged 2 to 6? It does not seem that most of them will be able to read the stories by themselves, so to which extent your system would be different to the tests carried out with caregivers reading the story? In my opinion, I do not see a very large market for personalized stories for children of this age, but I might be wrong.
- Could you please define the notions of learnability, imageability, and appropriatness as they were explained to the annotators?
- It is not clear for me how the % of Age-Appropriate Stories were assessed. How many experts did you used? Were they able to reach a good agreement?

**Reasons To Accept:**

- Three real contributions in a field that will become more and more important.
- Good paper, well-written and reporting solid work.

**Reasons To Reject:**

- The authors seems to consider that De Belder and Moens (2010) are deprecated technologies for LS, but they base their analysis of the text reading difficulty in readability formulae dating back more than 70 years, ignoring advances in the field (see Vajjala, 2021). In particular, these formulas measures the length of word rather than AoA or word frequency, which might not be the best choice in relation to paper’s aims.
- The different annotation processes are described in a rather naive way, with no sign of annotation guidelines, not computation of interrater agreements.

Vajjala, S. (2021). Trends, limitations and open challenges in automatic readability assessment research. arXiv preprint arXiv:2105.00973.

**Reproducibility:**

3: Could reproduce the results with some difficulty. The settings of parameters are underspecified or subjectively determined; the training/evaluation data are not widely available.

**Reviewer Confidence:**

4: Quite sure. I tried to check the important points carefully. It's unlikely, though conceivable, that I missed something that should affect my ratings.

**Typos Grammar Style And Presentation Improvements:**

- Several references are incomplete, in the bibliography. Please check them systematically.

l.152: Biemiller and Slomin (Biemiller and Slomin, 2001) → Biemiller and Slomin (2001)
l.578: which the at least → “the” should be suppressed.
l.628: models are unable generate →  models are unable to generate

---

> ### Author Rebuttal · Authors · 2023-08-29
>
> Thank you so much for your review! We think you characterize the main contributions of our paper very well, and we appreciate the thought and detail you put into reviewing our work. We also found your reasons to reject insightful and would like to address those here as well as in our paper in order to ameliorate any concerns.
>
> For one, you mention that our methodology includes readability metrics which are outdated and only capture aspects of the text such as word and sentence length, which we agree is not very conducive to the aims of our paper. The readability measures we used, although well-established and widely used, are coarse oversimplification of language use. We appreciate the inclusion of (Vajjala, 2021) for reference and have examined it for possible improvements to our current measurements. This paper mentions the lack of readily available tools or code as one of the main limitations to automatic readability assessment, which is something our own research has reflected as well. It cites just two studies that have code publicly available (Ambatti et al., 2016 and Howcroft & Demberg, 2017), one of which is for directly comparing readability between two samples as opposed to providing a metric for readability in general, and both of which can only assess readability at an individual sentence level. Since we are using several different measures consistently between the human- and computer-written stories, however, we believe these imperfect measures serve as a good starting point for comparison against the BfP corpus, which is a corpus of commercially available published picture books for children ages 5 and younger.
>
> Further, we also agree that our annotation process lacks description in this iteration of the paper and would like to include full definitions of learnability, imageability, and appropriateness as they were presented to our annotators in the appendix of our paper for future readers. Thank you for pointing this out! In regard to the percentage of age-appropriate stories, this was simply decided automatically by the stories’ words’ age of acquisition. We also would like to clarify that for each example to be added to our newly created CDS dataset, the top answer had to be agreed upon by 3 or more annotators, two of whom always included a professor and PhD student working in the field of Natural Language Processing.
>
> You also mention some questions regarding the intended use case for these stories. You are correct, the children that are the targets for these stories are preschoolers and are not independent readers. Thus, they will require an adult caregiver to read the stories to them. Caregivers reading stories to children, often called “book sharing”, is an important way in which children increase their vocabulary. Finding ways to increase a child’s vocabulary by including target words in contexts that appeal to the child’s current interests is one of the long-term goals of this project.
>
> Finally, thank you for all of the additional sources you have provided! These are great resources surrounding dyslexic children in relation to our topic and definitely will enable more exploration in that direction.

---

### Meta-Review · Area_Chair_pjf9 · 2023-09-15

**Recommendation:** 4

**Metareview:**

The reviewers agree that this is an interesting and well-motivated study, and while there were some concerns raised about the lack of certain comparisons/evaluations and a lack of detail about the annotation process, discussion with the authors helped resolve these areas of concern, for the most part. They presented some follow up results and discussion, which should be included in the next draft of the paper (particularly the experiments regarding the different stages of the pipelined approach).

---

### Decision · Program_Chairs · 2023-10-07

**Decision:**

Accept-Main

**Comment:**

The reviewers agree that this is an interesting and well-motivated study, and while there were some concerns raised about the lack of certain comparisons/evaluations and a lack of detail about the annotation process, discussion with the authors helped resolve these areas of concern, for the most part. They presented some follow up results and discussion, which should be included in the next draft of the paper (particularly the experiments regarding the different stages of the pipelined approach).